# Physical Activity, Sleep Patterns and Diet Habits as Well as the Prevalence of Obesity among Adolescents: A Cross Sectional Study from Ha’il City in Saudi Arabia

**DOI:** 10.3390/ijerph192316174

**Published:** 2022-12-02

**Authors:** Salma Abedelmalek, Halima Adam, Sultan Alardan, Sami Yassin, Hamdi Chtourou, Nizar Souissi

**Affiliations:** 1Department of Sport Science and Physical Activity, College of Education, University of Ha’il, Hail 55255, Saudi Arabia; 2Laboratory of Physiology and Functional Explorations, Faculty of Medicine, Sousse 4002, Tunisia; 3Department of Psychology, College of Education, University of Ha’il, Hail 55255, Saudi Arabia; 4Research Unit Physical Activity, Sport and Health (UR18JS01), National Observatory of Sports, Tunis 1003, Tunisia; 5High Institute of Sport and Physical Education of Sfax, University of Sfax, Sfax 3003, Tunisia; 6High Institute of Sport and Physical Education, Ksar-Saïd, Manouba University, Manouba 1003, Tunisia

**Keywords:** obesity, physical activity, sleep patterns, diet habits, adolescents

## Abstract

Obesity is expected to increase in the Kingdom of Saudi Arabia (KSA). Therefore, the objective of this study was to determine the physical activity, sleep patterns and diet habits as well as the prevalence of obesity among adolescents from Ha’il City. A cross-sectional study was conducted with 1598 adolescent students (663 males and 935 females) aged 12–15 years who were randomly selected from different schools in Ha’il. Body mass index (BMI) was used to classify participants as underweight, normal weight, overweight and obese (class and class II). Moreover, physical activity, sleep patterns and diet habits were recorded. The prevalence of obesity was 52.1%. Obesity is significantly more prevalent in females compared to males (62.9% vs. 36.59%, *p* < 0.001). Moreover, students in the third grade are the most obese in comparison with the first and the second educational level (*p* < 0.001). Additionally, students aged 15 years old are the most obese compared to 12 years, 13 years and 14 years (*p* < 0.001). Additionally, the results showed that students who suffer from obesity eat food when they feel higher stress and tension scores and that they do not eat their meals regularly. Additionally, 79% of those who eat fried food daily are obese. It was reported that 61.1% of students in this study were physically inactive. Furthermore, 58.4% of students have a higher proportion of insufficient sleeping duration (>7 h per night) (*p* < 0.001). This increased rate of obesity is higher in females compared to males and it was related to inactivity as well as perturbed sleep and eating habits. Therefore, we recommend an obesity prevention program including health education in Ha’il City, KSA.

## 1. Introduction

The prevalence of obesity has increased in many countries around the world, and it is predicted that by 2030, the majority of the world’s adult population will be obese or overweight (51% of the population will be obese) [1]. In 2016, according to the World Health Organization data, 340 million children and adolescents aged 5–18 years women and men were obese [1]. In recent years, obesity has reached pandemic proportions and the rising trend of obesity is directly influenced by a variety of factors (i.e., sleep disorder, hereditary, demographic factors, physical inactivity and bad food habits) [2,3]. In recent decades, obesity among children and adolescents is a major concern in Saudi Arabia (KSA). KSA is a country with one of the with the highest obesity and overweight prevalence rates, with 7 out of 10 people experiencing this problem [4]. According to the Global Burden of Diseases GBD (2015) [5], obesity prevalence in adults in the Eastern Mediterranean Region (EMR) increased from 15% in 1980 to 21% in 2015. The prevalence of obesity among KSA adults was 35.6% in 2000 [6]. Despite the prevalence of obesity among Saudi adolescents, there is a paucity of published research on the epidemiology of physical activity, food habits and sleep patterns and their link to obesity among Saudi teenagers [7]. This fact was highlighted in a review of sleep medicine in KSA, which emphasized the need for an investigation on sleep in the country, particularly to address the prevalence of various sleep disruptions among different populations [8]. Recently, it has been reported that overweight and obesity are related to sleep disorders in children [9]. Furthermore, overweight and obesity are correlated with physical inactivity and bad food habits. It is well documented that sleep deprivation is a substantial risk factor for obesity [10]. Previous reports discovered an inverse relationship between obesity and short sleep durations in Saudi teenagers, and weight gain is associated with an increased desire for unhealthy food [11,12]. It is well recognized that an accurate assessment of physical activity levels is an important factor for understanding the association between an active lifestyle and health [11]. In addition, physical inactivity is crucial for monitoring the prevalence of obesity in adolescents [12]. The prevalence of overweight and obesity in KSA is well documented, but there is an obvious lack of information on regional differences, prevalence data, and quality of life, which were not included in several studies.

Ha’il is a city in the eastern-border region of KSA. To our knowledge, no previous studies have been conducted on overweight and obesity in this region at any age. The only study conducted in this region was part of a study of 13 regions in KSA, which reported that 1023 (21.7%) from 4709 participants were obese in 13 regions in KSA, with 20.7% from a region of Ha’il [2]. As a result, the current study aimed to examine the sleep, physical activity and diet habits as well as the prevalence of obesity among teenagers aged 12 to 15 years in Ha’il city. We hypothesized that there are significant relationships between sleep duration and eating and physical activity habits, and markers of overweight and obesity.

## 2. Methods

### 2.1. Study Design and Setting

This is a cross-sectional study that was conducted over a two-month period (from January to March). After receiving ethical approval, all information related to the study was distributed to different intermediate schools via the education administration in Ha’il; a total of 26 schools were selected from different regions in the city. Anthropometric measurements were realized for all participants. Additionally, the participants had to respond to questionnaires that were related to the dependent variables.

### 2.2. Study Population and Sample

A cross-sectional study was conducted with 1598 students (663 males and 935 females) adolescents who were randomly selected from different schools in Ha’il, a city formed of approximately 597,144 inhabitants, lying in the eastern-border region of KSA. Students were selected according to their sex (i.e., male and female), educational level (first, second and third level) and age (from 12 to 15 years old) (Table 1). The minimum sample size was calculated, and the sample proportion was within 0.05 of the population with a 95% confidence level [13]. A total of 10% of the sample were excluded from the study because of missing data. A total of 26 classes were selected in the city (13 from each of the boy and girl schools). The study was open to all students in the selected classes, and they were recruited from schools. In addition, the schools’ consent as well as the students’ and their parents’ approval for conducting the survey were all secured. A written “Informed Consent” was obtained and signed by the child’s legal guardian and parents. In addition, the assent of a minor through the signature of a witness was obtained on the “Certificate of Assent of a Minor”.

Participants were promised that all data would be used exclusively for research purposes during the informed consent process. According to Google’s privacy policy assessed on 10 February 2022 (https://policies.google.com/privacy?hl=en), participants’ responses are anonymous and confidential. Participants might stop the study and leave the questionnaire at any point before the submission process. In fact, only the given “submit” button was used to save responses. By completing the survey, participants acknowledged their voluntary consent to participate. Participants were requested to be honest in their responses. The assumption of 5% for duplicate participants, entry errors and eligibility of inclusion and exclusion criteria provided a revised sample of 1598 participants.

### 2.3. Inclusion and Exclusion Criteria

The inclusion criteria were healthy (they did not present motor problems or injury) Saudi male and female students (12–15 years old) living in KSA at the time of the study. The health statuses of the students who accepted to participate were checked by medical records. A standardized measurement protocol was employed in all participating data collections to ensure accurate and consistent measurements throughout this multi-school project. A screening of participants’ health statuses and ages for eligibility against inclusion and exclusion criteria led to the exclusion of 6 participants with cognitive decline/impairment (identified by a medical record) and 19 participants aged > 15 years old.

### 2.4. Anthropometric Measurements

The anthropometric variables included body weight and height. Measurements were performed in the morning by a trained researcher according to written standardized procedures. Body weight was measured using a digital scale (Tanita Corporation, Tokyo, Japan, precision = ±0.1 kg). Height was measured to the nearest cm while the subject was in the full standing position (in the Frankfort horizontal plane) without shoes using a calibrated portable stadiometer. BMI was calculated as a ratio of weight in kg by height squared in meters according to the standards of the World Health Organization (WHO) to classify children as underweight (BMI ≤ 18.5 kg·m^2^); normal weight (18.5–24.9 kg m^2^); overweight (BMI 25–29.9 kg·m^2^) and obese [class I (30–34.9 kg·m^2^), class II (35–39.9) and class III (≥40 kg·m^2^)]. The index of central obesity (ICO), defined as a ratio of waist circumference (WC) and height, was used to identify overweight and obesity in adolescents aged between 14–17 years [14]. WC was measured horizontally with the subject standing, and was rounded off to the nearest 0.1 cm using a non-stretchable measuring tape at the level of the umbilicus and at the end of gentle expiration.

### 2.5. Survey and Outcome Measures

Questionnaire and anthropometric measurements were used for data collection. Designed electronic surveys were conducted to access the different sleep patterns, dietary habits and activity lifestyles. In addition, the daily time spent on video games and internet use [15] were recorded.

### 2.6. Self-Reported Dietary Habits Questionnaire

A self-reported diet questionnaire was used for data collection. The questionnaire was designed to record eating and food habits (i.e., number of meals/day, skipping breakfast daily and eating fast food weekly). Additionally, the daily eating of snacks and junk food as well as vegetables and fruits was recorded, as observed in previous studies [16,17], in parts referring to the Nutricalc questionnaire [18]. Nutritionists and professional translators performed linguistic validations to ensure that the translation from English to Arabic had the same intention as the questions from the source questionnaire. Subsequently, the electronic version of the developed survey and a pilot test with 20 participants were used to assess the implementation of the questionnaire and reliability of the measurements. The questionnaire was edited further, in light of results and feedback, and an improved version was developed and used.

### 2.7. International Physical Activity Questionnaire Short Form (IPAQ-SF)

The IPAQ-SF was used to obtain internationally comparable data on health-related physical activity. It is a 9-item scale, assessing the amount of minutes spent on vigorous and moderate intense activity and walking during the last 7 days. In addition, the amount of minutes spent sitting on weekdays during the past 7 days is assessed. Data from the IPAQ-SF are summed within each item (i.e., vigorous intensity, moderate intensity and walking) to estimate the total amount of time spent engaged in physical activity per week [14].

### 2.8. Sleep Patterns

Recent research suggests a relationship between a short sleep duration and obesity [13]. The section aims to assess the data related to the number of typical sleeping hours per day in adolescents using the self-reported questionnaire. In the present study, the Arab Teens Lifestyle Study (ATLS) research instrument used for the collection of lifestyle information consisted of 47 items, including the first five items that the researcher must measure/record [19]. Only sleep data are analyzed and presented. Insufficient sleep—sleeping less than 7 h per night—was defined according to the National Sleep Foundation for an adolescent population [20].

### 2.9. Data Analysis

Descriptive statistics were used to define the proportion of responses for each question and the total distribution in the total score of each questionnaire. All statistical analyses were performed using the IBM^®^ SPSS^®^ Statistics program (Paris, France, version 20.0). Normality of the data distribution was confirmed using the Shapiro–Wilks W-test. Values were computed and reported as mean ± SD (standard deviation) as well as frequency counts (%) for all variables. A t-test was used to determine the prevalence of obesity and the difference between the means of two groups (i.e., male and female). To identify the differences in body mass index according to two variables, a two-way analysis of variance ANOVA was used. The Bonferroni post hoc test was performed whenever significant effects or significant interactions were seen. The collinearity of data was checked using the bivariate Pearson’s product moment correlation (r) calculation. All statistical tests used were two-sided with a type I error (α) of 0.05.

### 2.10. Ethical Considerations

This study was reviewed and approved by the Research Ethics Committee at the University of Ha’il (Number: H-2021-219; Date: 12 June 2021).

## 3. Results

### 3.1. The Characteristics of the Study Sample

A total of 1598 adolescents students (663 males and 935 females)—who were randomly selected from different schools in Ha’il, taking into accordance their educational level (first, second and third level) and age (from 12 to 15)—participated in this study (Table 1).

### 3.2. The Prevalence of Obesity among the Sample

The results shown in Table 2 reported that 52.1% of the students are obese.

### 3.3. Differences in Body Mass Index According to Demographic Variables

Regarding the differences in students’ BMI according to demographic variables, the results reported that females are more obese than males (62.9% vs. 36.59%, *p* = 0.0001; Table 3). Additionally, 2.6% of females and 1.3% of males are obese class II. Furthermore, it was found that obesity (class I and II) is higher among third-grade students compared to the second grade and first grade (*p* = 0.0001; Table 3). According to age, it was found that the 15-year-old students are the most obese (Class I and Class II, *p* = 0.0001; Table 3).

### 3.4. Differences in Body Mass Index According to Health Variables

Differences in the body mass index according to the obesity of a family member and the central obesity index were summarized in Table 4. A total of 51% of the students answered yes to the question “Does a family member suffer from obesity?” and 478 (459 obese class I and 19 obese class II) of the obese students answered yes to this question, which indicates that a family member’s obesity contributes to the differences between students (*p* = 0.0001). Moreover, it was found that the obese students had a high central obesity index than the other BMI groups (*p* = 0.0001). The regression test showed that it is possible to predict obesity through the central obesity rate (R^2^ = 0.208, F = 419.852, *p* = 0.0001). The finding revealed that ICO positively predicts obesity (β = 0.456, *p* < *0*.0001). This result indicated that females who are obese class I and class II have more central obesity than males who are obese class I and class II [F(4,1593), = 29.10, *p* = 0.0001].

### 3.5. Differences in Body Mass Index According to Eating Habits

Data of the differences between BMI groups according to the eating habits are shown in Table 5. The results revealed that the students who answered “Yes” when asked if “they wake up at night to eat” are obese (559 class I, and 23 class II), and they represent approximately 63.3% of the total obese participants (*p* = 0.0001). The results showed that students who suffer from obesity eat food when they feel stress and tension (581 class I, and 24 class II; *p* = 0.0001). Additionally, from the 881 students who reported that they do not eat meals regularly, 515 are obese, (495 class I, 20 class II, *p* = 0.0001). Eating fried food is very popular among students—748 of them answered that they eat it 3 to 5 days a week (from them, 497 are obese). When testing the differences between students according to the number of times they eat fried food, it was found that 79% of those who eat it daily are obese (175 class I, 6 class II, Table 5).

### 3.6. Differences in Body Mass Index According to Physical Activity

Data related to the difference in BMI classification according to physical activity are shown in Table 6. A total of 977 (61.1%) students reported that they do not exercise, of whom 614 are obese (*p* = 0.0001). Differences were tested in the students’ answers regarding not participating in intense physical activities or exercising during weekdays—it was found that the students who answered never are the most obese among all groups (*p* = 0.027). When analyzing the data, it was found that 942 students, 58% of the sample members, do not engage in any moderate physical activity, and 559 are obese (*p* = 0.0001). It was reported that students who do not walk at all are the most obese, followed by those who walk once or twice a week (*p* = 0.0001).

### 3.7. Differences in Body Mass Index According to Sleep Patterns

The results revealed that 656 students slept less than seven hours, while 942 reported that they get enough sleep (more than 7 h). A total of 471 obese students slept less than 7 h (Table 7). Moreover, a significant correlation between BMI and sleep patterns (r = −0.294 *p* = *0*.0001) as well as physical activity level (r = 0.300, *p* = 0.0001) was observed.

## 4. Discussion

Obesity among children and adolescents has become a worldwide public health threat around the word [21]. The results of the present study indicated that obesity is prevalent among middle school students and is inevitably affected by a number of factors addressed in the study. Previous studies reported that obesity is prevalent among children aged 12–19 by 20.6%, who are the most obese, compared to all groups of children [22]. In the same context, it has been reported that 28.2% of the students in middle school are overweight and obese in the city of Buraidah, KSA [23]. The results of a study conducted in the Ha’il region revealed that obesity is prevalent by 63.6% [24].

Moreover, the gender of students has a prominent role in the differences in BMI, and obesity among students. The results showed that girls were more obese than boys in middle school. Previous results showed that female students in middle school complain of weight gain more than boys [25]. The prevalence of obesity among female students of school age was 20.9%, while among male students, was 20.4% [22]. In addition, women in Ha’il are 14.8% more obese than men [24]. Studies conducted in various countries such as India, the US, and others indicate an increase in body mass in women [26,27,28]. Due to the different nature of the physical composition of girls, they are more likely to gain weight than boys are. Moreover, the intermediate stage is a transitional stage in which students leave primary school seats and move from childhood to adolescence. Thus, it was noted that students in the third grade, who are the oldest among the sample members, are the most ghee. Researchers believe that adolescence has a significant impact on students gaining more weight due to psychological changes and a decrease in self-regulation skills that limit the ability to control impulsive behaviors, thus, leading to an inability to control eating [29]. Some adolescents appear to have a higher BMI from eating disorders, as well as emotional and subjective difficulties [30].

Among the striking results in the current study, it was found that 57% of students whose family member suffers from obesity are also affected. Researchers attribute this finding to parenting patterns that contribute to children gaining excess weight. Overweight parents influence the eating behavior of their offspring from an early age [31]. The role of genetic factors in obesity, and the relationship of obesity in fathers with obesity in children, has been well demonstrated [32]. The results of the study showed that the rate of central obesity predicts weight gain. After examining the study participants’ data, it became clear that obese women have a higher index of central obesity than men do. The increase in central obesity is one of the most critical risk factors for obesity and is associated with many health risks [33]. Studies have also recorded a significant increase in the central obesity rate, which is expected to reach 38% in boys and 4% in girls by 2030 [28].

In the present study, the results revealed that students who wake up at night to eat are the most obese, and they feel stress and tension. A significant association exists between obesity and stress. This is in agreement with a previous study [34] reporting that stress-related eating is highly prevalent among 16-year-old girls and is associated with obesity. In the present study, we focused on the frequency of food intake rather than the quantity, due to methodological reasons (i.e., the method is relatively simple and time-efficient). Despite the single item in the assessment of stress-related eating, the study revealed the feasibility of the results associated with obesity. A previous study reported that obesity is associated with adverse social, educational and psychological factors [35]. Other factors (unfortunately not included in the analyses) associated with stress-related eating seem to be a crucial determinant of children’s obesity (i.e., home availability of foods, eating styles) [36,37]. In this context, 79.4% of those who eat fried food daily are obese. This agrees with a previous study [38], indicating that obese adolescents choose energy-dense food. This is in line with previous data, which found a significant association between obesity and more frequent fast-food consumption, due to its high fat and calories [39]. In the present study, there was a significant association between obesity and excessive use of PlayStation and social media. In fact, the findings reported that most of the participants who are obese play PlayStation between 3 to 5 h. For the participants who are obese, social media screens decrease energy disbursement by spending less time on physical activity and increasing their consumption of obesogenic foods. This is in agreement with a previous study [40]. Thus, the time spent playing electronic games is enough to increase calorie consumption and reduce the metabolic rate. Eating while watching TV is also a common practice among families. Additionally, both males and females who are obese reported stress-related eating, more frequent meals and a shorter sleep duration (i.e., less than 7 h). Indeed, we observed a significant correlation between BMI, sleep patterns and physical activity. Furthermore, the higher percentage of the students with obesity in the current study were physically inactive. This percentage of obesity increases remarkably with a lack of exercise practice (i.e., intense and moderate physical activities) or a decrease in time practicing exercise per week. Both males and females who are obese did not engage in 10 min of walking.

## 5. Study Limitation

A strength of this research project is the inclusion of different schools in the Ha’il region from KSA. Additionally, this study investigated the prevalence of obesity according to different factors related to obesity (e.g., sleep and dietary habits, participation to physical activity, etc.). However, the current study presents some limitations. Firstly, we focused only on the frequency of food intake rather than quantity. Additionally, the cross-sectional design of this study could be a limitation. The validity of answers is a general problem of online surveys, although we requested participants to be honest in their responses, as described in the methods section.

## 6. Conclusions

We concluded that obesity is prevalent in Ha’il, KSA, and is a serious public health problem. Obesity and overweight were high among adolescent female students, and were strongly related to physical activity, sleep patterns and diet habits among adolescents. Thus, we recommend adolescent students to control obesity, and to practice regular exercise as well as health and nutritional education. We expect to use the results to develop physical and nutritional programs.

## Figures and Tables

**Table 1 ijerph-19-16174-t001:** The characteristics of the study sample (*n* = 1598).

		*n*	%	Mean	SD
Gender	Males	935	55.8	28.89	4.89
	Females	663	39.6	27.28	4.62
Educational level	First	216	12.9	24.54	5.35
	Second	496	29.6	27.59	5.39
	Third	886	52.9	29.48	3.75
Age	12	110	17.7	24.56	5.13
	13	335	27.7	25.91	5.71
	14	733	34	28.38	4.09
	15	420	19	30.75	3.62

*n*: number of participants; %: percentage; SD: standard deviation.

**Table 2 ijerph-19-16174-t002:** Body mass index classification among the sample of the study.

BMI Classification	*n*	%	Mean	SD	85th Percentile	95th Percentile
Under weight	85	5.3	17.25	0.743	17.89	18.06
Normal	164	10.3	20.62	2.64	24.35	24.84
Overweight	517	32.4	26.26	0.82	27.16	27.83
Obese (class I)	798	49.9	31.88	1.29	33.41	34.2
Obese (class II)	34	2.1	36.43	0.98	37.71	38.41
Obese (class III)	0	0	0	0	-	-

*n*: number of participants; %: percentage; SD: standard deviation.

**Table 3 ijerph-19-16174-t003:** The distribution (in %) for body mass index according to the demographic characteristics (i.e., sex, educational level and age).

			Under Weight	Normal	Over-Weight	Obese (Class I)	Obese (Class II)
Sex	Female	*n*/%	48/5.13	88/9.41	210/22.45	564/60.30	25/2.6
		Mean	17.12	20.49	26.25	31.87	36.28
		SD	0.797	2.543	0.794	1.31	0.806
	Male	*n*/%	37/5.58	76/11.46	307/46.30	234/35.29	9/1.3
		Mean	17.43	20.77	26.27	31.91	36.86
		SD	0.635	2.769	0.843	1.26	1.317
Educational level	First	*n*/%	29/13.42	62/28.7	77/35.64	48/22.22	-
		Mean	16.85	20.23	26.34	31.84	-
		SD	0.897	2.242	0.946	1.325	-
	Second	*n*/%	44/8.8	62/12.5	139/28.1	243/47.17	8/1.6
		Mean	17.45	19.78	26.3	31.85	36.95
		SD	0.571	1.71	0.867	1.313	1.313
	Third	*n*/%	12/1.35	40/4.51	301/33.9	507/57.22	26/2.82
		Mean	17.52	22.52	26.22	31.89	36.28
		SD	0.494	3.429	0.766	1.286	0.821
Age (year)	12	*n*/%	11/10	34/30.9	44/40	21/19.9	-
		Mean	16.53	20.27	26.32	32.03	-
		SD	0.906	2.146	0.876	1.332	-
	13	*n*/%	42/12.53	68/20.29	104/31.1	118/35.2	3/0.89
		Mean	17.34	19.75	26.24	31.92	37.63
		SD	0.716	1.952	0.873	1.342	1.237
	14	*n*/%	25/3.4	47/6.41	315/42.9	332/45.2	14/1.9
		Mean	17.35	22.05	26.24	31.81	36.18
		SD	0.567	3.026	0.787	1.261	0.935
	15	*n*/%	7/1.6	15/3.57	54/12.8	327/77.8	17/4.04
		Mean	17.5	20.88	26.34	31.92	36.43
		SD	0.658	3.383	0.895	1.312	0.867

*n*: number of participants; %: percentage; SD: standard deviation.

**Table 4 ijerph-19-16174-t004:** Differences in body mass index according to health variables.

			Under Weight	Normal	Over-Weight	Obese I	Obese II
A family member is obese	Yes	*n*/%	32/3.9	57/7.08	238/29.56	459/57.01	19/0.23
		mean ± SD	17.38 ± 0.677	21.18 ± 3.28	26.28 ± 0.81	31.85 ± 1.30	36.69 ± 1.08
	No	*n*/%	53/6.6	107/13.4	279/35.18	339/42.74	15/1.89
		mean ± SD	17.17 ± 0.776	20.31 ± 2.19	26.24 ± 0.83	31.91 ± 1.28	36.12 ± 0.75
Index of central obesity	Non central obesity	*n*/%	79/9.32	149/17.59	350/38.9	258/30.46	11/1.2
		mean ± SD	17.25 ± 0.752	20.43 ± 2.49	26.22 ± 0.771	31.81 ± 1.30	36.96 ± 1.15
	Central obesity	*n*/%	6/0.7	15/1.76	167/19.6	540/63.4	23/2.7
		mean ± SD	17.26 ± 0.673	22.49 ± 3.41	26.33 ± 0.919	31.91 ± 1.29	36.18 ± 0.794
PlayStation play	Less than 1 h	*n*/%	18/7	39/23.3	109/42.4	88/34.2	3/1.1
		mean ± SD	17.22 ± 0.857	19.91 ± 1.78	26.19 ± 0.71	31.74 ± 1.31	36.09 ± 0.87
	1 h	*n*/%	12/7.1	19/11.3	72/43.1	63/37.7	1/0.6
		mean ± SD	17.05 ± 0.762	21.55 ± 2.70	26.14 ± 0.73	31.44 ± 1.19	36.44 ± 0.0
	2 h	*n*/%	44/17.12	91/9.1	269/6.9	564/56.6	27/2.7
		mean ± SD	17.34 ± 0.676	20.69 ± 2.77	26.34 ± 0.87	31.97 ± 1.31	36.53 ± 1.03
	3 to 5 h	*n*/%	8/5.8	10/7.29	49/35.9	67/48.9	3/2.18
		mean ± SD	17.21 ± 0.809	21.24 ± 3.82	26.07 ± 0.816	31.74 ± 1.17	35.90 ± 0.64
	More than 5 h	*n*/%	3/7.1	5/11.9	18/42.8	16/23.8	-
		mean ± SD	17.01 ± 1.03	20.13 ± 2.44	26.51 ± 0.86	31.82 ± 1.18	
Social media	Do not use	*n*/%	16/13.23	31/25.61	60/49.51	13/10.74	1/10.82
		mean ± SD	17.39 ± 0.74	20.03 ± 1.97	26.15 ± 0.74	31.86 ± 1.27	36.44 ± 0.0
	1 h	*n*/%	11/7.38	22/10.83	84/41.3	85/41.87	1/0.4
		mean ± SD	16.88 ± 0.89	19.93 ± 1.77	26.21 ± 0.72	31.63 ± 1.32	35.62 ± 0.0
	2 h	*n*/%	54/4.8	101/9.04	344/30.7	589/52.7	29/2.5
		mean ± SD	17.28 ± 71	20.93 ± 2.86	26.33 ± 0.86	31.60 ± 1.29	36.45 ± 0.99
	3 to 5 h	*n*/%	3/2.56	8/6.83	21/17.9	82/70	3/2.56
		mean ± SD	17.62 ± 28	19.93 ± 1.46	25.87 ± 0.60	31.72 ± 1.21	36.57 ± 1.22
	More than 5 h	*n*/%	1/2.5	2/20	8/20	29/72.5	-
		mean ± SD	16.52 ± 0.0	24.36 ± 7.85	25.73 ± 0.68	31.72 ± 1.28	

*n*: number of participants; %: percentage; SD: standard deviation.

**Table 5 ijerph-19-16174-t005:** Differences in body mass index according to eating habits.

			Under Weight	Normal	Over-Weight	Obese I	Obese II
Wake up to eat at night	Yes	*n*/%	32/3.48	75/8.16	229/24.9	559/60.8	23/2.5
		mean ± SD	17.46 ± 0.615	20.59 ± 2.839	26.25 ± 0.81	31.88 ± 1.30	36.39 ± 1.03
	No	*n*/%	53/5.7	89/13.08	288/42.35	239/35.14	11/1.6
		mean ± SD	17.13 ± 0.79	20.64 ± 2.488	26.27 ± 0.83	31.87 ± 1.32	36.52 ± 0.90
I eat when I feel stressed	Yes	*n*/%	35/3.6	74/7.6	258/26.54	581/59.7	24/22.01
		mean ± SD	17.49 ± 0.54	20.93 ± 14.3	26.23 ± 0.79	31.88 ± 1.30	36.33 ± 1.03
	No	*n*/%	507.9	90/11.3	259/41.1	217/34.4	10/1.58
		mean ± SD	17.09 ± 0.82	20.37 ± 2.20	26.28 ± 0.85	31.87 ± 1.27	36.68 ± 0.83
Eat meals regularly	Yes	*n*/%	52/7.25	80/11.15	268/37.3	303/42.2	14/1.9
		mean ± SD	17.39 ± 0.74	20.03 ± 1.97	26.15 ± 0.74	31.86 ± 1.27	36.44 ± 0.0
	No	*n*/%	33/3.7	84/9.53	249/28.26	495/56.1	20/2.27
		mean ± SD	16.88 ± 0.89	19.93 ± 1.77	26.26 ± 0.81	31.88 ± 1.28	36.69 ± 1.04
I eat fried food	Rarely	*n*/%	33/27.78	62/13.05	227/47.8	146/30.7	7/1.4
		mean ± SD	17.29 ± 0.72	20.95 ± 2.73	26.22 ± 0.75	31.65 ± 1.25	36.39 ± 1.06
	1 to 2	*n*/%	16/10.95	35/23.9	9464.38	-	1/0.68
		mean ± SD	17.11 ± 0.72	20.39 ± 2.09	26.19 ± 0.86		37.54 ± 0.0
	3 to 5	*n*/%	29/3.87	53/7.08	169/22.59	477/63.7	20/2.6
		mean ± SD	17.36 ± 0.66	20.60 ± 2.78	26.37 ± 0.88	31.92 ± 1.28	36.49 ± 1.05
	Daily	*n*/%	7/3.05	14/6.11	27/11.7	175/76.4	6/2.62
		mean ± SD	16.95 ± 1.17	19.82 ± 2.95	26.16 ± 0.79	31.95 ± 1.34	36.11 ± 0.64

*n*: number of participants; %: percentage; SD: standard deviation.

**Table 6 ijerph-19-16174-t006:** Differences in body mass index according to physical activity.

			Under Weight	Normal	Over-Weight	Obese I	Obese II
Exercise	Yes	*n*/%	50/0.80	101/16.26	252/40.5	213/34.3	5/0.8
		mean ± SD	17.19 ± 0.78	20.58 ± 2.48	26.34 ± 0.84	31.83 ± 1.21	36.47 ± 0.86
	No	*n*/%	35/3.28	63/6.44	265/2.12	585/59.8	29/2.96
		mean ± SD	17.35 ± 0.68	20.69 ± 2.90	26.18 ± 0.79	31.90 ± 1.32	36.43 ± 1.01
Intense physical activities	No	*n*/%	51/5.01	105/23.3	296/29.13	542/53.5	22/8.7
		mean ± SD	17.27 ± 0.72	20.84 ± 2.78	26.26 ± 0.83	31.91 ± 1.28	36.61 ± 1.04
	1 to 3	*n*/%	22/5.9	36/11.3	143/38.6	161/43.5	8/2.6
		mean ± SD	17.12 ± 0.86	20.48 ± 2.68	26.33 ± 0.86	31.76 ± 1.32	35.97 ± 0.86
	4 to 5	*n*/%	4/2.72	10/7.1	49/41.8	54/46.1	-
		mean ± SD	17.67 ± 0.47	19.90 ± 1.63	20.10 ± 0.68	31.65 ± 1.20	
	>5	*n*/%	8/8.42	13/7.1	29/7.8	41/43.1	4/13.4
		mean ± SD	17.31 ± 0.65	19.75 ± 1.84	26.11 ± 0.68	32.16 ± 1.34	36.39 ± 0.62
Moderate physical activity	No	*n*/%	47/4.98	80/8.94	256/27.1	536/56.9	23/2.4
		mean ± SD	17.28 ± 0.73	20.90 ± 2.87	26.27 ± 0.82	31.96 ± 1.29	36.54 ± 1.07
	1 to 3	*n*/%	23/5.3	55/12.8	171/39.9	170/39.7	9/2.1
		mean ± SD	17.20 ± 0.72	20.39 ± 2.27	26.30 ± 0.86	31.67 ± 1.29	36.16 ± 0.75
	4 to 5	*n*/%	7/4.86	16/11.11	58/40.2	62/43.05	1/0.6
		mean ± SD	17.37 ± 0.80	21.11 ± 3.19	26.13 ± 0.71	31.65 ± 1.21	35.80 ± 0.0
	>5	*n*/%	8/9.52	13/15.4	32/38.09	30/35.7	1/1.19
		mean ± SD	17.14 ± 0.91	19.27 ± 1.21	26.16 ± 0.70	32.09 ± 1.40	37.11 ± 0.0
Walk for ten minutes	I did not walk	*n*/%	12/2.37	17/3.3	50/9.9	403/79.8.7	23/4.55
		mean ± SD	17.43 ± 0.83	21.18 ± 3.62	26.20 ± 0.82	32.0 ± 1.31	36.46 ± 1.08
	1 to 2 days	*n*/%	36/8.08	51/11.46	199/41.7	154/34.6	5/1.12
		mean ± SD	16.94 ± 0.86	20.32 ± 2.47	26.26 ± 0.86	31.79 ± 1.25	36.36 ± 0.65
	3 to 5 days	*n*/%	12/3.75	46/14.37	130/40.6	129/39.3	3/0.93
		mean ± SD	17.47 ± 0.34	21.07 ± 2.43	26.33 ± 0.81	31.76 ± 1.25	36.42 ± 0.75
	>5 days	*n*/%	25/7.6	50/15.24	138/42.07	112/34.1	3/0.91
		mean ± SD	17.52 ± 0.45	20.32 ± 2.61	26.22 ± 0.81	31.68 ±1.26	36.39 ± 1.20

*n*: number of participants; %: percentage; SD: standard deviation.

**Table 7 ijerph-19-16174-t007:** Differences in BMI according to sleep patterns.

		Under Weight	Normal	Over-Weight	Obese I	Obese II
Less than 7 h	*n*	22	36	127	450	21
	Mean	17.27	21.86	26.23	31.9	36.56
	SD	0.72	3.52	0.83	1.31	1.06
More than 7 h	*n*	63	128	390	348	13
	Mean	17.25	20.27	26.27	31.85	36.22
	SD	0.75	2.24	0.81	1.26	82

*n*: number of participants; %: percentage; SD: standard deviation.

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
