# Peer review of "Physical Activity, Sleep Patterns and Diet Habits as Well as the Prevalence of Obesity among Adolescents: A Cross Sectional Study from Ha’il City in Saudi Arabia"

_ijerph, 2022, doi:10.3390/ijerph192316174_

Round 1

Reviewer 1 Report

This paper reports obesity for n=1598 12-15 year-olds from the  Ha’il City in Saudi Arabia.  The paper is in general hard to follow and assesses obesity inappropriately (in my view).  Major revisions are required.

It is inappropriate to use adult obesity cutoffs for 12-15 year-olds.  The WHO defines obesity according to BMI-for age growth standards for 5-19 year-olds.  This (or similar) should be used.  It is inappropriate to apply the same obesity standards to differently aged children (e.g., 12 and 15 year-olds) as has been done, and it is inappropriate to conclude from the results that obesity increases with age, as Table 1 seems to imply.

In general the English needs to be improved, as do the design and content of the tables.

Some specific comments.

1. Abstract: How are male and female obesity prevalences both lower than the overall prevalence?

2. What are the means in table 1, 2, 4, 5, 7, 8 & 9?  If they are BMIs, how can the the means for the overweight group be<25 (if overweight is defined as 25-29.99)?  Moreover, I do not think that within-group BMIs are very interesting (tables 2 onward), given these are necessarily constrained by the categorizations of the groups.

3. Tables should report the mean BMI across characteristics (or proportion in obesity groups across characteristics) and indicate which are significant (the test statistic and p).  The full details of ANOVAs (or whatever test has been conducted) are not necessary.  

4. Why is figure 3 presented before figure 2, and where is figure 4?

5. I can't follow what the ANOVAs are testing in tables 4, 5, 7, 8 & 9.  Is the ANOVA being conducted on the BMI or obesity group membership?  If the latter, ANOVA is inappropriate - chi-square or a logistic regression would be more sensible.  

Author Response

We thank the reviewers and the editor for their thorough review of our work and for the very constructive and helpful comments. We have considered the comments and have provided specific responses for each remark. Our responses appear in red typeface. We hope that this version has been improved and that is now suitable for publication in your journal. Furthermore, we are ready to make any further changes that would be deemed necessary for any deeper improvement.

Reviewer 1

This paper reports obesity for n=1598 12-15 year-olds from the  Ha’il City in Saudi Arabia.  The paper is in general hard to follow and assesses obesity inappropriately (in my view).  Major revisions are required.

We thank again the reviewer for their thorough review of our work and for the constructive and helpful comments. We hope that this version has been improved and that it is now suitable for publication in your journal.

It is inappropriate to use adult obesity cutoffs for 12-15 year-olds. The WHO defines obesity according to BMI-for age growth standards for 5-19 year-olds.  This (or similar) should be used.  It is inappropriate to apply the same obesity standards to differently aged children (e.g., 12 and 15 year-olds) as has been done, and it is inappropriate to conclude from the results that obesity increases with age, as Table 1 seems to imply.

In general the English needs to be improved, as do the design and content of the tables.

 The reviewer is right. Corrections made as suggested. We have seriously worked on the quality of the English language.

Although, the WHO defines a BMI classification for different age groups, we suggest that the used classification is appropriate and suitable to conclude BMI according to the different age groups.

Please see the new version. We hope that this version has been improved.

Some specific comments.

  • Abstract: How are male and female obesity prevalences both lower than the overall prevalence?

This was corrected and the BMI classification was changed.

  • What are the means in table 1, 2, 4, 5, 7, 8 & 9?  If they are BMIs, how can the the means for the overweight group be<25 (if overweight is defined as 25-29.99)?  Moreover, I do not think that within-group BMIs are very interesting (tables 2 onward), given these are necessarily constrained by the categorizations of the groups.

The data was represented in clearer manner. We hope that this version of Tables is better.

  • Tables should report the mean BMI across characteristics (or proportion in obesity groups across characteristics) and indicate which are significant (the test statistic and p).  The full details of ANOVAs (or whatever test has been conducted) are not necessary.  

The reviewer is right, these detail were removed in the revised version.

  • Why is figure 3 presented before figure 2, and where is figure 4?

The reviewer is right. These figures were removed. Only tables were used in the revised version.

  1. I can't follow what the ANOVAs are testing in tables 4, 5, 7, 8 & 9.  Is the ANOVA being conducted on the BMI or obesity group membership?  If the latter, ANOVA is inappropriate - chi-square or a logistic regression would be more sensible.  

The new results according to the new classification are presented in the new version. We present new tables in the revised version.

Reviewer 2 Report

Dear Authors,

I would like to note the relevance of the research and its importance. A better understanding of the factors affecting the level of physical activity and the level of obesity among young people is the basis for the development of effective preventive programs and the basis for political decisions. In turn, all this is necessary for the prosperity of the nation.

I would like to make only a few remarks that will allow me to correct some inaccuracies in the article.

1. Better-quality figures should be added (they are currently unclear in the manuscript). Also, all figures do not have (*) marks, which allows comparing data

2. For all used questionnaires, it is worth adding information (with relevant links) confirming the validity of these questionnaires for this group of respondents.

Kind regards,

Author Response

We thank the reviewers and the editor for their thorough review of our work and for the very constructive and helpful comments. We have considered the comments and have provided specific responses for each remark. Our responses appear in red typeface. We hope that this version has been improved and that is now suitable for publication in your journal. Furthermore, we are ready to make any further changes that would be deemed necessary for any deeper improvement.

Reviewer 2

I would like to note the relevance of the research and its importance. A better understanding of the factors affecting the level of physical activity and the level of obesity among young people is the basis for the development of effective preventive programs and the basis for political decisions. In turn, all this is necessary for the prosperity of the nation.

I would like to make only a few remarks that will allow me to correct some inaccuracies in the article.

  • Better-quality figures should be added (they are currently unclear in the manuscript). Also, all figures do not have (*) marks, which allows comparing data

The reviewer is right. In the revised version, we used only tables to present the data.

  1. For all used questionnaires, it is worth adding information (with relevant links) confirming the validity of these questionnaires for this group of respondents.

Correction made as suggested. Please see changes made in the revised version.

Reviewer 3 Report

In my opinion the paper can be informative and provide a valuable source document for anyone requiring a primer to know and understand this issue. But, numerous shortcomings in the section Methods and Results make this paper not appropriate for publication in this form and significant corrections should be made (major revision). Some comments:      

  • Line 19: Add study method.
  • Line 40: Instead of the cited reference No 2, the appropriate reference should be correctly inscribed. The cited reference No 2 contains the appropriate reference that was cited in that paper (by WHO). 
  • Lines 47-49: Add the appropriate reference for the GBD 2015 study.  
  • Line 65: Emphasize and explain why it is important to conduct this and similar research in those aged 12-15 years. 
  • Lines 74-157: Reconstruct the stated text, consolidate in the section Methods. I assume that `Data analysis` in this manuscript should belong to the section Methods, see the Instructions to authors. According to the Instructions to authors, correctly mark all subsections. The text should be presented more clearly in subsections `Study setting`, `Study design`, `Study population`, `Study sample`, `Study sample calculation`, ... 
  • Line 78: If the results presented in Table 1 are the results of this manuscript, place Table 1 in the section Results. Describe in detail data presented in Table 1 in the section Results. In the section Methods, all variables presented in this manuscript should be defined (e.g. define `Education level` shown on Table 1, etc.). 
  • Line 83: State the exact period (from ... to) for the two months when this study was realized. 
  • Lines 91-92: State the reference number of the permission of the Ethical Committee and the date when this approval was granted. Especially emphasize whether the approval for participation in the study was written, informed and voluntary. State how the approval was obtained from the parents. 
  • Line 92: State the way how the participants were recruited for this study. 
  • Lines 95-99: Define the term `healthy`. Explain how it was determined in this manuscript that study participants were `healthy`. 
  • Lines 95-99: The `Exclusion Criteria` are not listed.  
  • Line 95-99: State the `Participant rate` and `Response rate` in this study. 
  • Lines 106-107: State the classification of the Body Mass Index, with an appropriate reference. 
  • Lines 122-128: State the results of the assessment of psychometric characteristics of the Arabic version of the stated questionnaire, with a citation of the reference where those results were published. 
  • Lines 130-131: Cite an appropriate reference for the mentioned questionnaire. 
  • Lines 130-134: Describe the mentioned questionnaire in an appropriate manner. 
  • Lines 131-134: State the results of the assessment of psychometric characteristics of the Arabic version of the stated questionnaire, with a citation of the reference where those results were published.     
  • Line 141: Define the abbreviation `ATLS`. 
  • Lines 141-143: Describe the used `the ATLS research instrument` in an appropriate manner. 
  • Line 144: state the results of the assessment of psychometric characteristics of the Arabic version of the stated questionnaire, with a citation of the reference where those results were published.   
  • Lines 145-157: Apart from the stated, define all statistical parameters that will be presented in the paper for the assessment of statistical significance.  
  • Line 158: Completely reconstruct the section Results, so that all results are presented and described in an appropriate order. 
  • Lines 148-152: In the entire section Results, in text and in all Tables, align and reduce the number of decimals, so that the results are more clear. This remark is not applied to the values of probability. 
  • Lines 160-163: The values for probability are completely inconsistently stated in the paper, starting from Table 1 and onwards, with a lack of alignment to how it is stated in the text where the description belonging to Tables is stated. E.g.,  `p` is written sometimes as `p = 0.0001]` sometimes as `P` as `.000`, see Table 1 then onwards. One manuscript should present values for probability in the same manner, as for all other results (that is, choose one nomenclature from multiple existing ones, but stick to it throughout the entire paper). It must be clear that it is not good practice as it is presented in Table 1 and in the description of this Table 1: it is not the same to write `0.0001` or `.000`. The remark goes to the entire section Results (Lines 158-242). 
  • Line 164: Add a new Table that will describe the distribution (in %) for Body Mass Index according to the already stated demographic characteristics of the study participants, i.e. by sex, educational level and age. 
  • Lines 177-179: There is a complete confusion with the description of Figure 3 and other Figures: the order of Figures, their titles, citation in the text of the Results, description of Figures. All Figures in this manuscript are insufficiently correctly put together, e.g., which is the most important, names of the horizontal and vertical axis are not states, values on the axes are not stated, etc.   
  • Line 192: Figure 4 can not be found in this manuscript. 
  • Lines 284-285: Explain which were the `methodological reasons` in this manuscript. 
  • Lines 298-299: It is not correct to cite the results of other studies in the way as done in this sentence, i.e. as it says `This is in agreement with the study conducted by 298 [40].`. This way was used in multiple places in the text of the entire manuscript: revise, correct and cite correctly. 
  • Lines 309-310: Cite Table 10 in text of the section Results, with a description of the data . 
  • Line 316: Explain whether your previous statement (Lines 284-285: `In the present study, we focused on frequencies of food intake rather than quantities, due to methodological reasons.`) can or cannot cause another limitation of this study.   
  • Line 316: Discuss the design of the study (cross-sectional study design) as a limitation of this study, firstly in the context of the results.  

Author Response

            We thank the reviewers and the editor for their thorough review of our work and for the very constructive and helpful comments. We have considered the comments and have provided specific responses for each remark. Our responses appear in red typeface. We hope that this version has been improved and that is now suitable for publication in your journal. Furthermore, we are ready to make any further changes that would be deemed necessary for any deeper improvement.

Reviewer 3

In my opinion the paper can be informative and provide a valuable source document for anyone requiring a primer to know and understand this issue. But, numerous shortcomings in the section Methods and Results make this paper not appropriate for publication in this form and significant corrections should be made (major revision). Some comments:      

We thank again the reviewer for their thorough review of our work and for the very constructive and helpful comments. We hope that this version has been improved and that is now suitable for publication in your journal.

Line 19: Add study method.

Correction made as suggested.

Line 40: Instead of the cited reference No 2, the appropriate reference should be correctly inscribed. The cited reference No 2 contains the appropriate reference that was cited in that paper (by WHO). 

The reviewer is right. Sorry for the lack of precision. Correction made as suggested.

Lines 47-49: Add the appropriate reference for the GBD 2015 study.  

Correction made as suggested by the reviewer.

Line 65: Emphasize and explain why it is important to conduct this and similar research in those aged 12-15 years. 

The reviewer is right. Corrections made as suggested. Please see the new version.

Lines 74-157: Reconstruct the stated text, consolidate in the section Methods. I assume that `Data analysis` in this manuscript should belong to the section Methods, see the Instructions to authors. According to the Instructions to authors, correctly mark all subsections. The text should be presented more clearly in subsections `Study setting`, `Study design`, `Study population`, `Study sample`, `Study sample calculation`, ... 

Correction made as suggested by the reviewer and respect the guidelines.

Line 78: If the results presented in Table 1 are the results of this manuscript, place Table 1 in the section Results. Describe in detail data presented in Table 1 in the section Results. In the section Methods, all variables presented in this manuscript should be defined (e.g. define `Education level` shown on Table 1, etc.). 

The Table 1 is about the characteristics of the subjects participated in the study. So, we suggest it could be presented in the methods section.

Line 83: State the exact period (from ... to) for the two months when this study was realized. 

Correction made as suggested by the reviewer.

Lines 91-92: State the reference number of the permission of the Ethical Committee and the date when this approval was granted. Especially emphasize whether the approval for participation in the study was written, informed and voluntary. State how the approval was obtained from the parents. 

Correction made as suggested by the reviewer. Information about the permission of the Ethical Committee was added. Please see the new version.

Line 92: State the way how the participants were recruited for this study. 

Correction made as suggested by the reviewer.

Lines 95-99: Define the term `healthy`. Explain how it was determined in this manuscript that study participants were `healthy`. 

Correction made as suggested by the reviewer.

Lines 95-99: The `Exclusion Criteria` are not listed.  

The reviewer is right. Corrections made as suggested. We added more information about the exclusion criteria of the participants. Please see the new version.

Line 95-99: State the `Participant rate` and `Response rate` in this study. 

Correction made as suggested by the reviewer.

Lines 106-107: State the classification of the Body Mass Index, with an appropriate reference. 

Correction made as suggested by the reviewer.

Lines 130-131: Cite an appropriate reference for the mentioned questionnaire. 

The reviewer is right. An appropriate reference was added. Please see the new version.

Lines 130-134: Describe the mentioned questionnaire in an appropriate manner. 

Correction made as suggested by the reviewer.

Lines 131-134: State the results of the assessment of psychometric characteristics of the Arabic version of the stated questionnaire, with a citation of the reference where those results were published.     

Correction made as suggested by the reviewer. Reference was added

Line 141: Define the abbreviation `ATLS`. 

Correction made as suggested by the reviewer. Please see the new version.

Lines 141-143: Describe the used `the ATLS research instrument` in an appropriate manner. 

Correction made as suggested by the reviewer. Please see the new version.

Line 144: state the results of the assessment of psychometric characteristics of the Arabic version of the stated questionnaire, with a citation of the reference where those results were published.   

Correction made as suggested by the reviewer. Reference was added.

Lines 145-157: Apart from the stated, define all statistical parameters that will be presented in the paper for the assessment of statistical significance.  

Correction made as suggested by the reviewer. Please see the new version.

Line 158: Completely reconstruct the section Results, so that all results are presented and described in an appropriate order. 

This section was rewritten. Please see the revised version.

Lines 148-152: In the entire section Results, in text and in all Tables, align and reduce the number of decimals, so that the results are more clear. This remark is not applied to the values of probability. 

Correction made as suggested by the reviewer. Please see the new version.

Lines 160-163: The values for probability are completely inconsistently stated in the paper, starting from Table 1 and onwards, with a lack of alignment to how it is stated in the text where the description belonging to Tables is stated. E.g.,  `p` is written sometimes as `p = 0.0001]` sometimes as `P` as `.000`, see Table 1 then onwards. One manuscript should present values for probability in the same manner, as for all other results (that is, choose one nomenclature from multiple existing ones, but stick to it throughout the entire paper). It must be clear that it is not good practice as it is presented in Table 1 and in the description of this Table 1: it is not the same to write `0.0001` or `.000`. The remark goes to the entire section Results (Lines 158-242). 

The reviewer is right. Sorry for the lack of precision. We have present values for probability in the same manner. Correction made as suggested.

Line 164: Add a new Table that will describe the distribution (in %) for Body Mass Index according to the already stated demographic characteristics of the study participants, i.e. by sex, educational level and age. 

Correction made as suggested by the reviewer.

Lines 177-179: There is a complete confusion with the description of Figure 3 and other Figures: the order of Figures, their titles, citation in the text of the Results, description of Figures. All Figures in this manuscript are insufficiently correctly put together, e.g., which is the most important, names of the horizontal and vertical axis are not states, values on the axes are not stated, etc.   

The figures were removed in the new version. Please see the new version.

Line 192: Figure 4 can not be found in this manuscript. 

Correction made as suggested by the reviewer. Please see the new version.

Lines 284-285: Explain which were the `methodological reasons` in this manuscript. 

Correction made as suggested by the reviewer. In the present study, we focused on frequencies of food intake rather than quantities, due to methodological reasons (i.e., the method relatively simple and time-efficient manner). Please see the new version.

Lines 298-299: It is not correct to cite the results of other studies in the way as done in this sentence, i.e. as it says `This is in agreement with the study conducted by 298 [40].`. This way was used in multiple places in the text of the entire manuscript: revise, correct and cite correctly. 

Correction made as suggested by the reviewer.

Lines 309-310: Cite Table 10 in text of the section Results, with a description of the data . 

This tables was removed. Please see the revised version.

Line 316: Explain whether your previous statement (Lines 284-285: `In the present study, we focused on frequencies of food intake rather than quantities, due to methodological reasons.`) can or cannot cause another limitation of this study.   

Correction made as suggested by the reviewer.

Line 316: Discuss the design of the study (cross-sectional study design) as a limitation of this study, firstly in the context of the results.  

Correction made as suggested by the reviewer.

Round 2

Reviewer 1 Report

I thank the authors for their thorough reworking of the paper.  All the issues I raised in my initial review have now been addressed.  I just have three minor comments/suggestions:

1. Lines 125-6. Presumably "according to the standards of the World Health Organization (WHO)" means "classified according to the age- and sex-specific standards of the World Health Organization (WHO)"?  If that is the case, perhaps say so.

2. Table 2: Signify that these are WHO categories and, if possible, indicate the percentile ranges in a column next to the 'BMI classification' column.  E.g., is underweight <2nd pecentile, and overweight between the 85th and 95th percentile?

3. Table 3. 82% of 15yo in obesity class II & III seems incredibly high.  Can these results be checked?

Author Response

We thank the reviewers and the editor for their thorough review of our work and for the very constructive and helpful comments. We have considered the comments and have provided specific responses for each remark. Our responses appear in red typeface. We hope that this version has been improved and that is now suitable for publication in your journal.

Reviewer 1

I thank the authors for their thorough reworking of the paper.  All the issues I raised in my initial review have now been addressed.  I just have three minor comments/suggestions:

  1. Lines 125-6. Presumably "according to the standards of the World Health Organization (WHO)" means "classified according to the age- and sex-specific standards of the World Health Organization (WHO)"?  If that is the case, perhaps say so.

More detail related to the BMI classification were added. Please see changes made in the revised version (L.121-125).

  1. Table 2: Signify that these are WHO categories and, if possible, indicate the percentile ranges in a column next to the 'BMI classification' column.  E.g., is underweight <2nd pecentile, and overweight between the 85th and 95th percentile?

We added the percentile ranges as suggested by the reviewer. Please see the new version (Table 2).

 Table 3. 82% of 15yo in obesity class II & III seems incredibly high. Can these results be checked?

Checked. Please see changes made in the revised version (Table 3).

Reviewer 3 Report

Thank you for the invitation to review the revised version of this paper. It was very time consuming and strenuous to check what was done by the authors, because in the response letter the authors have simply stated that they answered the comment and I had to go back and forth and try to find where that was done in the revised manuscript, instead of either giving the lines or sections where the changes can be seen or even citing the new information in the response letter. However - this showed that on more than a few occasions the changes were not made even though stated so. This is misleading. The following remarks remain:  

- Line 39: My comment on the previous version of manuscript indicated that the authors cited a reference which mentioned the BMI cut-off values and cited the appropriate reference. Instead of citing a paper that cites it - the authors need to cite the original reference. In the revised version authors cited ref. 1, which is not appropriate for this. Check the section Methods and reference No. 15 in the paper you have previously cited here (Althumiri NA, Basyouni MH, AlMousa N, AlJuwaysim MF, Almubark RA, BinDhim NF, Alkhamaali Z, Alqahtani SA. Obesity in Saudi Arabia in 2020: Prevalence, Distribution, and Its Current Association with Various Health Conditions. Healthcare (Basel). 2021 Mar 11;9(3):311. doi: 10.3390/healthcare9030311. PMID: 33799725; PMCID: PMC7999834.).

- The previously made comment for subsections in the section Methods still stands. Use the logical order when describing the study - describe the study design, study setting, study population and sample, measured variables and instruments used for it, data analysis, ethical considerations. Currently subheadings are not presented in a coherent and logical manner. - Results section should not be 5.0 subheading - check and revise. -  Once again, Table 1 presents characteristics of the participants. If these are the results of this study, the data to which you came by collecting information from participants etc. - it does not belong to the section Methods. - Where can the definition of variables be seen? The authors did not answer this comment. Define all variables in the section Methods, e.g. educational level that you mention in Table 1. -  Comment was: "Especially emphasize whether the approval for participation in the study was written, informed and voluntary. State how the approval was obtained from the parents." The authors stated that this information was added. Where? - How were the participants recruited, not only where, explain the process. - Lines 116-117: The authors still did not explain how they defined "healthy". How was this screening process done? Interview, medical exam, check of medical records? - Lines 116-117: How did you measure and define cognitive decline/impairment in these children? - Even though authors stated they added, they did not state exclusion criteria in the revised paper. - Authors said they did but they did not add - participation and response rate. - Etc - see the original comments. Provide a detailed response letter that will cite the change that was made or provide rationale for no change.

Author Response

We thank the reviewers and the editor for their thorough review of our work and for the very constructive and helpful comments. We have considered the comments and have provided specific responses for each remark. Our responses appear in red typeface. We hope that this version has been improved and that is now suitable for publication in your journal.

Reviewer 3

Thank you for the invitation to review the revised version of this paper. It was very time consuming and strenuous to check what was done by the authors, because in the response letter the authors have simply stated that they answered the comment and I had to go back and forth and try to find where that was done in the revised manuscript, instead of either giving the lines or sections where the changes can be seen or even citing the new information in the response letter. However - this showed that on more than a few occasions the changes were not made even though stated so. This is misleading. The following remarks remain:  

- Line 39: My comment on the previous version of manuscript indicated that the authors cited a reference which mentioned the BMI cut-off values and cited the appropriate reference. Instead of citing a paper that cites it - the authors need to cite the original reference. In the revised version authors cited ref. 1, which is not appropriate for this.

The reviewer is right. We changed the reference by the following one:

  • Finkelstein, E. A.; Khavjou, O. A.; Thompson, H.; Trogdon, J. G.; Pan, L.; Sherry, B.; & Dietz, W. Obesity and severe obesity forecasts through 2030. American journal of preventive medicine.2012, 42(6), 563-570.

Please see change made in the revised version (L. 39).

Check the section Methods and reference No. 15 in the paper you have previously cited here (Althumiri NA, Basyouni MH, AlMousa N, AlJuwaysim MF, Almubark RA, BinDhim NF, Alkhamaali Z, Alqahtani SA. Obesity in Saudi Arabia in 2020: Prevalence, Distribution, and Its Current Association with Various Health Conditions. Healthcare (Basel). 2021 Mar 11;9(3):311. doi: 10.3390/healthcare9030311. PMID: 33799725; PMCID: PMC7999834.).

This was checked. The appropriate reference is ;

[15] Parikh, R. M., Joshi, S. R., Menon, P. S., & Shah, N. S. Index of central obesity–A novel parameter. Medical hypotheses. 2007, 68, 1272-1275.

Please see change made in the revised version (L. 127).

- The previously made comment for subsections in the section Methods still stands. Use the logical order when describing the study - describe the study design, study setting, study population and sample, measured variables and instruments used for it, data analysis, ethical considerations. Currently subheadings are not presented in a coherent and logical manner. - Results section should not be 5.0 subheading - check and revise.

Correction made as suggested. Please see change made in the revised version (methods section).

-  Once again, Table 1 presents characteristics of the participants. If these are the results of this study, the data to which you came by collecting information from participants etc. - it does not belong to the section Methods.

Correction made as suggested. Please see change made in the revised version (results section).

- Where can the definition of variables be seen? The authors did not answer this comment. Define all variables in the section Methods, e.g. educational level that you mention in Table 1.

Correction made as suggested. Please see change made in the revised version (L.87-88).

-  Comment was: "Especially emphasize whether the approval for participation in the study was written, informed and voluntary. State how the approval was obtained from the parents." The authors stated that this information was added. Where?

Correction made as suggested. Please see change made in the revised version (L.94-96).

- How were the participants recruited, not only where, explain the process.

All students of the schools in Hail city were included. And then according to the exclusion criteria or absence or refusal to participate, they were not included to the study.

- Lines 116-117: The authors still did not explain how they defined "healthy". How was this screening process done? Interview, medical exam, check of medical records?

The health status of students who accepted to participate was checked by a medical records. Healthy for this study is related to that participants didn’t present motor problems or injury.

- Lines 116-117: How did you measure and define cognitive decline/impairment in these children?

The cognitive decline/impairment was verified by a medical record. Please see changes made in the revised version (L.112-114).

- Even though authors stated they added, they did not state exclusion criteria in the revised paper.

These information are presented in the revised version (L.107-114).

- Authors said they did but they did not add - participation and response rate. - Etc - see the original comments. Provide a detailed response letter that will cite the change that was made or provide rationale for no change.

We suggest that we responded to all comments with the specification of the sections/lines. Please see change made in the revised version. Otherwise, we are ready to respond or to made the required changes.
